# Sequential Immunizations with Influenza Neuraminidase Protein Followed by Peptide Nanoclusters Induce Heterologous Protection

**DOI:** 10.3390/v16010077

**Published:** 2024-01-03

**Authors:** Wen-Wen Song, Mu-Yang Wan, Jia-Yue She, Shi-Long Zhao, De-Jian Liu, Hai-Yan Chang, Lei Deng

**Affiliations:** 1Hunan Provincial Key Laboratory of Medical Virology, College of Biology, Hunan University, Changsha 410082, China; songwwen@outlook.com (W.-W.S.); wanmuyang@hnu.edu.cn (M.-Y.W.); shejiayue@hnu.edu.cn (J.-Y.S.); slzhao@hnu.edu.cn (S.-L.Z.); dejianliu@hnu.edu.cn (D.-J.L.); 2College of Life Sciences, Hunan Normal University, Changsha 410082, China; 3Beijing Weimiao Biotechnology Co., Ltd., Haidian District, Beijing 100093, China

**Keywords:** influenza vaccine, neuraminidase, serum cross-reactivity, cross-protection, nanocluster, sequential immunization

## Abstract

Enhancing cross-protections against diverse influenza viruses is desired for influenza vaccinations. Neuraminidase (NA)-specific antibody responses have been found to independently correlate with a broader influenza protection spectrum. Here, we report a sequential immunization regimen that includes priming with NA protein followed by boosting with peptide nanoclusters, with which targeted enhancement of antibody responses in BALB/c mice to certain cross-protective B-cell epitopes of NA was achieved. The nanoclusters were fabricated via desolvation with absolute ethanol and were only composed of composite peptides. Unlike KLH conjugates, peptide nanoclusters would not induce influenza-unrelated immunity. We found that the incorporation of a hemagglutinin peptide of H2-d class II restriction into the composite peptides could be beneficial in enhancing the NA peptide-specific antibody response. Of note, boosters with N2 peptide nanoclusters induced stronger serum cross-reactivities to heterologous N2 and even heterosubtypic N7 and N9 than triple immunizations with the prototype recombinant tetrameric (rt) N2. The mouse challenge experiments with HK68 H3N2 also demonstrated the strong effectiveness of the peptide nanocluster boosters in conferring heterologous protection.

## 1. Introduction

As the second most abundant glycoprotein on the surface of influenza A virion envelope, NA plays several roles in viral replication, most crucially in removing terminal sialic acid from glycan receptors and enabling the release of viral progeny. However, the latent capacity of NA-specific cross-protection potency was previously underestimated in the evaluation of influenza vaccine effectiveness. In recent years, mounting experimental evidence has demonstrated the protectiveness of NA immunity against multiple influenza A viruses of diverse HA subtypes, indicating the potential of NA antigen incorporation for improving current seasonal influenza vaccines or even developing a universal influenza vaccine [1,2]. With an increasing number of isolated NA-specific monoclonal antibodies (mAbs), many cross-reactive B-cell epitopes in the NA head domain have been identified and characterized [3,4,5]. However, it is so far still greatly difficult to efficiently enhance the antibody responses against these specific B-cell epitopes via vaccinations with NA protein immunogens.

Due to their occluded positions or a low affinity to precursor B-cell receptors, certain cross-reactive B-cell epitopes of NA often exhibit subdominant immunogenicity in the immunodominance hierarchies after influenza vaccinations or infections [6,7]. To improve this suboptimal cross-protection, we developed a peptide-based sequential immunization regimen to promote immuno-focusing by precisely steering the bulk immune responses toward the elicitation of antibody specificities at a selected region where these peptides originate from [8]. We further verified that the NA protein imprint is the prerequisite for effective peptide-based boosters. This idea was inspired by previous designs of chimeric HA vaccines. Earlier studies achieving influenza cross-protection enhancement via reactivations of memory B cells have been documented, and sequential immunizations with chimeric HAs comprised of the exotic head domain and the conserved stalk domain of interest efficiently enhanced the stalk-specific cross-protective antibody responses against pan-group influenza strains and even the pandemic strains as well [9,10,11,12]. In comparison, due to the extremely low immunogenicity of peptide haptens, we still need to select a protein carrier or delivery vector to present peptide antigen for inducing an antibody response of a protective magnitude.

Conjugation of peptides with keyhole limpet hemocyanin (KLH) would inevitably strongly induce influenza-unrelated immunity against the protein carrier. Our successful experience in fabricating a desolvated peptide nanocluster suggests its potential technical application for the preparation of NA peptide nanoclusters. Ethanol desolvation drives the self-assembling of polypeptides, and this process should mainly be mediated by the spontaneous interactions among exposed hydrophobic residues; thus, the formation of nanoclusters does not require additional support from any scaffold or vector [13,14]. To ensure the induction of an NA peptide-specific antibody response, we designed composite peptides by fusing a peptide derived from the B-cell epitope of NA and a peptide derived from the T helper epitope of HA [15,16]. Here, we investigated if the incorporation of composite NA peptides into nanoclusters could retain an appreciative level of immunogenicity for antibody induction and boost immuno-protection.

## 2. Materials and Methods

### 2.1. Ethics Statement

All mouse experiments were performed in strict accordance with the animal welfare requirements of Hunan University. All mouse experimental protocols were approved by the Animal Ethics Committee of the College of Biology, Hunan University (Ethics File Code: HNUBIO202202001). The BALB/c mouse infection experiments using mouse-adapted H3N2 virus were performed in an animal facility that meets the biosafety level 2 requirement of Hunan Normal University.

### 2.2. Cell Lines

Spodoptera frugiperda 9 (Sf9) insect cells and human embryonic kidney 293T cells were purchased from the American Type Culture Collection (ATCC). Sf9 insect cells were cultured in a serum-free SF900 II medium (Suzhou world-medium Biotechnology Co., Ltd., Suzhou, China) supplemented with 100 mg/mL streptomycin and 100 U/mL penicillin (Beijing Solarbio Science & Technology Co., Ltd., Beijing, China) in sterile flasks at 27 °C at a speed of 110 revolutions per minute (rpm). The 293T cell line was cultured in Dulbecco’s Modified Eagle’s Medium (DMEM; Gibco, Grand Island, NY, USA) containing 10% fetal bovine serum (FBS; Gibco, Grand Island, NY, USA) at 37 °C, with 5% CO_2_.

### 2.3. Expression and Purification of Recombinant rtNA Proteins

The constructions of rtN1 and rtN2 were the same as previously reported [8]. The amino acid sequences of the NA head domains of rtN1 and rtN2 were derived from the NA sequences of influenza A trains A/California/04-CIP100_RGCM_1868/2009 (H1N1) N1 (GenBank Protein Accession: AWZ60965.1) and A/Kansas/14/2017 (KS17, H3N2) N2 (GenBank Protein Accession: QIA50738.1), respectively. Their encoding genes were codon optimized for preferential expressions in Sf9 insect cells and synthesized using an open reading frame in pFastbac-1 plasmid by Tsingke Biotechnology Co., Ltd., Changsha, China. The recombinant bacmids encoding rtN1 and rtN2 were generated by using the bac-to-bac baculovirus expression system for secretory expressions of rtN1 and rtN2 proteins, respectively. The Sf9 cell cultures were transfected with the recombinant bacmids to express recombinant baculoviruses. At seventy-two hours (h) after transfection, the supernatant containing recombinant baculoviruses was added to the Sf9 suspension cell cultures for infection for 72 h, and then the supernatant was collected and centrifuged to remove insect cells and cellular debris. The recombinant proteins from the collected supernatant were affinity-purified by using immobilized-metal affinity chromatography with Ni Sepharose 6 Fast Flow resin (Cytiva, Marlborough, MA, USA). In the elution step, rtN1 or rtN2 proteins bound to the purification column were eluted by using elution Tris buffers with stepwise gradients from 10 to 300 mM of imidazole. The eluted protein samples were concentrated in the Tris buffer (20 mM of Tris, 150 mM of NaCl, pH 7.8; Sangon Biotech (Shanghai) Co., Ltd. Shanghai, China) by using an ultrafiltration tube 30,000 NMWL (Merck Millipore, Billerica, MA, USA). Next, rtNA proteins were further purified by using size-exclusion chromatography (SEC) with a Superdex 200 Increase 10/300 GL column (Cytiva, Marlborough, MA, USA).

### 2.4. Preparation of Polypeptide Nanoclusters

The selection of N1P1 and N2P2 polypeptides was based on the conserved sequences of N1 and N2. We selected the peptides of the mouse CD4^+^ T-cell epitopes from influenza H3, including HA2(166–180) (amino acid sequence: ALNNRFQIKGVELKS) and HA1(183–199) (amino acid sequence: HHPSTNQEQTSLYVQAS), and chemically synthesized the composite peptides by fusing an NA peptide and an H3 peptide, with a flexible Ser-Gly-Gly-Ser (SGGS) linker (Shanghai Taopu Biotechnology Co., Ltd., Shanghai, China) in between. The peptide solution was desolvated by dripping anhydrous ethanol at a constant speed of 1 mL/min to 5 volumes of the peptide solution. The mixture was stirred at 200 rpm for approximately 20 min at room temperature until the solution turned turbid. Then, the peptide nanoclusters were pelleted via centrifugation at a speed of 12,000 rpm for 15 min at room temperature. The resulting nanoparticles were stored at 4 °C for later vaccination use. The flow chart of the fabrication process shown in Figure 2B was drawn using the online program Figdraw (www.figdraw.com; accessed on 5 November 2023).

### 2.5. Immunization and Viral Challenge Experiments

Specific-pathogen-free (SPF) 6–8-week-old female BALB/c mice were purchased from Hunan SJA Laboratory Animal Co., Ltd., Changsha, China. The immunogens of 3 μg of rtN1, 3 μg of rtN2, and 30 μg of KLH (AAT Bioquest, Sunnyvale, CA, USA) conjugates with peptides, or 30 μg of peptide nanoclusters in 50 μL of sterile phosphate-buffered saline (PBS, 137 mM of NaCl, 2.7 mM of KCl, 10mM of Na_2_HPO_4_, 1.8 mM of KH_2_PO_4_; Sangon Biotech (Shanghai) Co., Ltd. Shanghai, China), were emulsified with 50 μL of incomplete Freund’s adjuvant (Sigma-Aldrich, St. Louis, MO, USA). The BALB/c mice (*n* = 6 per group) were intraperitoneally immunized three times in three-week intervals. In the sequential immunization groups, mice were primed with rtNA and boosted twice with KLH-peptide conjugates or peptide nanoclusters. The immunization groups with triple immunizations with rtNA were the positive controls. The groups with boosters of unconjugated KLH and triple intraperitoneal injections with equal volumes of sterile PBS for mock immunization were the negative controls. Blood was collected one day prior to prime immunization and two weeks after each immunization. Three weeks after the last immunization, the mice were anesthetized via intraperitoneal injections with 1% sodium pentobarbital and then intranasally infected with 6 × median lethal dose (mLD_50_) of A/Hong Kong/8/1968 (HK68, H3N2) containing 600 plaque-forming units in 50 μL of sterile PBS. Body weight changes and survival rates were recorded on the day of intranasal infection and on day 5 post infection. At the end of the challenge experiment, the mice were euthanized by neck dislocation. The blood samples were incubated at 37 °C for 15 min, followed by centrifugation at 3000× *g* for 10 min, and then the serum was collected and stored at −20 °C.

### 2.6. Enzyme-Linked Immunosorbent Assay (ELISA)

ELISA immunoplates (JET BIOFIL, Guangzhou, China) were coated with 200 ng of rtNA protein antigens per well in 0.05 M of sodium bicarbonate buffer (pH 9.6), and incubated overnight at 4 °C. On the second day, the immunoplates were washed three times with PBS, and then 300 μL of the blocking buffer containing 5% skim milk in PBS supplemented with 0.5% Tween 20 (PBST) was added per well to block the immunoplates for 1 h at room temperature. The serum samples were 3-fold serially diluted in a buffer containing 50% skim milk solution and 50% PBST, starting from a dilution factor of 100, and then were added to the immunoplates for 1 h incubation at 37 °C for binding to antigens. After washing three times with PBST, the immunoplates were incubated with the horseradish peroxidase (HRP)-conjugated goat anti-mouse IgG antibody (ABclonal Technology Co., Ltd., Wuhan, China) for 1 h incubation at room temperature. Then, the immunoplates were washed with PBST three times, 50 µL of 3,3′,5,5′-Tetramethylbenzidine (TMB; Beijing Solarbio Science & Technology Co., Ltd., Beijing, China) was added per well, and the immunoplates were incubated at 37 °C for 15 min for catalyzing subtracts. Finally, 50 μL of 2 M H_2_SO_4_ was added per well to stop the reaction. Absorbance was read at 450 nm using a microplate reader (DLJ-200, DLJ Bio-Tech, Nanjing, China).

### 2.7. Cell-Based ELISA

The recombinant pCDNA3.1 plasmid (Tsingke Biotechnology Co., Ltd., Changsha, China) encoding NA from A/Japan/305/1957 (JP57) H2N2, A/Philippines/2/1982 (PH82) H3N2, A/Netherlands/219/2003 (NL03) H7N7, or A/Shanghai/02/2013 (SH13) H7N9 was transfected into the HEK293T cell culture. At twenty-four hours after transfection, the digested cells were seeded at a density of 6 × 10^4^ per well in 96-well cell culture plates that had been pre-treated overnight with poly-L-lysine for cell culture adhesion at 37 °C with 5% CO_2_. On the next day, the cells in the 96-well plates were fixed with 4% paraformaldehyde for 15 min at room temperature and washed twice with PBS, and the subsequent detection steps were described in the ELISA method.

### 2.8. Statistical Analysis

Unpaired *t*-test analysis was performed using GraphPad Prism 8. *p*-values less than or equal to 0.05 were considered significant. *, **, ***, and **** represent *p*-values less than 0.05, 0.01, 0.005, and 0.001, respectively. The abbreviation ns indicates not significant.

## 3. Results

### 3.1. Preparation and Characterization of Recombinant NA Proteins and Peptide Nanoclusters

The polypeptides of the globular head domains of N1 and N2 from A/California/04-CIP100_RGCM_1868/2009 (H1N1) and A/Kansas/14/2017 (H3N2) strains, respectively, were linked to the C-terminal of the tetrabrachion motif polypeptide with a flexible SGGS linker in between. The melittin signal peptide, hexahistidine tag, and enterokinase cleavage site were fused at the N-terminal of rtNA polypeptide for the purposes of secretory expression, affinity purification, and purification tag removal, respectively (Figure 1A). The protein bands of rtN1 and rtN2 showed the expected sizes in the Western blot analysis and Coomassie blue-stained acrylamide gel (Figure 1B,C). The rtN1 and rtN2 proteins were further purified by using the SEC column (Figure 1D), and their overall native antigenicity was well retained, as demonstrated by the strong bindings with anti-N1 and anti-N2 mouse serum, respectively (Figure 1E,F).

Self-assembled peptide nanoclusters have a remarkable advantage over KLH conjugates in avoiding the elicitation of influenza-irrelevant immune responses. To compensate for the loss of T helper epitopes provided by KLH, a conserved T helper epitope peptide from influenza H3 was incorporated into the composite peptide (Figure 2A). The way to fabricate peptide nanoclusters is depicted in Figure 2B. With an increasing amount of anhydrous ethanol added into the peptide solution, the liquid mixture turned turbid with stirring, suggesting the massive formation of self-assembled nanoparticles. The dynamic light scattering (DLS) analysis showed a normally distributed size range of N2P2TP2 nanoclusters and the peak of distribution was near the size of 100 nm.

### 3.2. Immunizations with Peptide Nanoclusters Enhanced NA-Specific Cross-Reactive Immune Response

To evaluate the serum cross-reactivity and protection potency of the resulting nanoclusters, mice that were primed with rtNA immunogen were further boosted twice with nanoclusters or peptide-KLH conjugates (Figure 3A). The hyperimmune serum from the groups of rtN1_(×3)_ and rtN2_(×3)_ showed the strongest binding activity to the homologous NA (Figure 3B,C). Boost immunizations with nanoclusters or peptide-KLH conjugates elicited varying degrees of enhancement in NA-specific antibody response, compared with the group of mice only receiving one prime immunization with rtNA (Figure 3B,C). The cross-reactivities of N2-specific serum were further evaluated via cell-based ELISA. Among all groups of mice with elicited N2 immunity, rtN2/N2P2TP1nano_(×2)_ induced the strongest serum reactivities to heterologous N2 from H2N2 and Phi H3N2, and heterosubtypic N7 and N9 (Figure 3D–G). The serum cross-reactivities of the group rtN2_(×3)_ were unexpectedly not much stronger than the group rtN2/KLH_(×2)_ (Figure 3D–G). Targeted induction of antibody response by peptide-based sequential immunizations may not produce as high an NA protein-specific IgG titer as the group rtNA_(×3)_, but it does show its superiority in enhancing cross-reactivities to NA of other subtypes in our study.

### 3.3. Peptide Nanoclusters Elicited Cross-Protection against Heterologous H3N2

Prophylaxis potency was evaluated in mouse challenge studies (Figure 3A). Three weeks after the second boost immunization, we used 6 × mLD_50_ of mouse-adapted HK68 H3N2 for intranasal challenge. On day 5 post infection, the body weight loss in the groups of rtN2_(×3)_, rtN2/N2P2-KLH_(×2)_, and rtN2/N2P2TP2nano_(×2)_ were comparable and significantly slower than that of the other groups (Figure 4). The significantly higher body weights in the groups of rtN2/N2P2-KLH_(×2)_ and rtN2/N2P2TP2nano_(×2)_ compared with the rtN2/KLH_(×2)_ group demonstrated the effectiveness of boosters with N2P2 peptide-based vaccines in targeted inductions of antibody response against B-cell epitopes where N2P2 originates from.

## 4. Discussion

Enhancement of cross-protection through influenza vaccinations can be achieved via targeted induction of antibody responses against conserved B-cell epitopes [8,17]. In this study, we investigated the effectiveness of NA peptide nanoclusters in enhancing cross-protective antibody responses and found that boost immunizations with nanoclusters elicited stronger serum cross-reactivities against NA proteins of diverse subtypes, indicating a greater potential to confer broader cross-protection against influenza A viruses. As elucidated in previous studies, only certain B-cell epitopes from NA are responsible for the elicitation of cross-protection [8,18]. Despite multiple effective boost immunizations with peptide immunogen, the titers of specific antibodies to the targeted B-cell epitopes were still not substantially increased for various reasons, like the presence of an idiotypic network that could play a role in controlling specific antibody responses [19]. Overall, this study provides a proof of concept for the design of NA peptide nanocluster immunogens that are effective in inducing cross-protective antibody responses.

Compared with conventional vaccines, the new generation of nanoparticle vaccines shows huge potential in terms of immunogenicity and antigen presentation. At present, a variety of nanoparticles have been found to be applicable for vaccine preparation, including inorganic nanoparticles, polymer nanoparticles, virus-like particles, and self-assembled protein nanoparticles [14,20,21]. We developed and characterized NA peptide nanoclusters without the loss of antigenicity and immunogenicity [13]. Ethanol desolvation triggers peptide self-assembly by reducing the solvent permittivity and polarity and drives the formation of nanoclusters. These desolvated nanoclusters are almost entirely composed of the peptide antigen of interest, which confers remarkable advantages. Firstly, the repetitive peptide antigen display and high antigen load facilitate the recognition and activation of specific precursor B cells, thereby enabling the effective inductions of antibody response. Secondly, the immunogenic component does not induce additional influenza-irrelevant immunities. With the aid of the H3 peptide-specific activation of CD4^+^ T helper response, all peptide nanoclusters were able to realize targeted antibody inductions (Figure 3B–G). The challenge experiment results showed that the vaccine effectiveness of the boosters with N2P2TP2 nanoclusters was as strong as that of peptide-KLH conjugates (Figure 4). Of note, the desolvation nanoparticle technology has been more widely applied and more successful in the field of chemical drug delivery. The nanoparticle albumin-bound strategy has allowed the creation of some effective anti-cancer nanoparticle drugs that are available on the market, like Abraxane nab-paclitaxel developed by Abraxis BioScience and Fyarro nab-sirolimus developed by Aadi Bioscience. Additionally, other gelatin-based and fibroin-based nanoparticles that can be prepared by using desolvation have also attracted considerable research interests regarding their applications in drug delivery, vaccine formulation, and biocatalysis [22,23].

The influenza-specific T lymphocyte response significantly contributes to the clearance of the influenza virus in both experimental animal models and humans [24,25,26]. Therefore, the design of a vaccine that induces T-cell cross-protection for single use or supplemental use with seasonal influenza vaccines could be a favorable option in the development of next-generation influenza vaccines. At present, some developed peptide-based vaccine candidates provide influenza cross-protection by inducing T-cell immunity and have been tested in clinical evaluations, such as Flu-V and M-001 [27,28,29]. We designed multi-epitope composite peptides comprising a B-cell epitope of NA and a T helper epitope of HA (Figure 2A). It was reported that the conserved H3 peptide HA2(166–180) of MHC II class restriction that was incorporated in N1P2TP and N2P2TP2 was able to provide adequate T-cell help and increase the antibody response against the co-delivered B-cell determinant peptide [30]. Mouse immunization experiments indicated the versatility of the conserved H3 peptide HA1(183–199) in inducing specific antibodies, T helper response, and cytotoxic T-cell response, and showed partial protections in mice against lethal-dose infection with X31 influenza virus [15]. We speculate that the incorporated HA peptides in our study must have provided adequate assistance in inducing NA peptide-specific antibody responses and might also elicit additional specific T-cell protection.

Our results demonstrated the superiority of peptide-based sequential immunizations in inducing broader serum cross-reactivity. In the serum antibody titration assays, the serum from the rtN2_(×3)_ group showed stronger binding activity to rtN2 protein than other groups (Figure 3C). On the contrary, the serum samples from the groups of rtN2/N2P2-KLH_(×2)_, rtN2/N2P2TP1nano_(×2)_, and rtN2/N2P2TP2nano_(×2)_ were able to show stronger binding activities when titrated against heterologous and heterosubtypic NA using cell-based ELISA (Figure 3D–G). In the challenge experiments, on day 5 post lethal-dose infection with the heterologous HK68 H3N2, the mice from the groups of rtN2/N2P2-KLH_(×2)_ and rtN2/N2P2TP2nano_(×2)_ experienced slight body weight losses comparable to the group of rtN2_(×3)_ (Figure 4), suggesting the peptide boosters are effective in enhancing protective antibody response. We did not observe better protections in the peptide booster immunization groups compared to the rtN2_(×3)_ group. In fact, the multiple aspects impacting the induced heterologous protection efficacy include the variations in the viral NA antigenicity and the magnitude of cross-reactive antibody response, which still needs to be further investigated to elucidate the main cause. The potential correlation of the variations in amino acid sequences with immunogenicity impairment was analyzed and discussed herein. We aligned the amino acid sequences of the NA proteins examined in this study (Figure 5) and found that the N2P2 peptide is highly conserved among all N2 proteins examined in this study, with only one site mutation K267P existing in the N2 proteins of the HK68, PH82, and JP57 strains. Only the corresponding peptide from SH13 N9 bears four site mutations compared with N2P2, which probably results in a reduction in the binding activity of N2P2-boosted serum samples compared to the heterosubtypic N9. As shown in the results of serum bindings to N9 in Figure 3G, the OD values (450 nm) in the groups of rtN2/N2P2-KLH_(×2)_ and rtN2/N2P2TP2nano_(×2)_ were not significantly higher than that of the rtN2/KLH_(×2)_ group. In this study, the immunization experiment employed only a female mouse model, as humoral immune responses to immunization and infection are generally stronger than male models [31,32]. It would be interesting to comprehensively evaluate peptide-based vaccine effectiveness in both female and male mouse models in the future.

## 5. Conclusions

Based on the progress in our recent work, we further explored our peptide nanocluster vaccine design and evaluated its immunogenicity and vaccine protectivity. The developed NA peptide nanocluster immunogen effectively boosted the cross-protective antibody responses by targeting cross-reactive B-cell epitopes in NA.

## Figures and Tables

**Figure 1 viruses-16-00077-f001:**
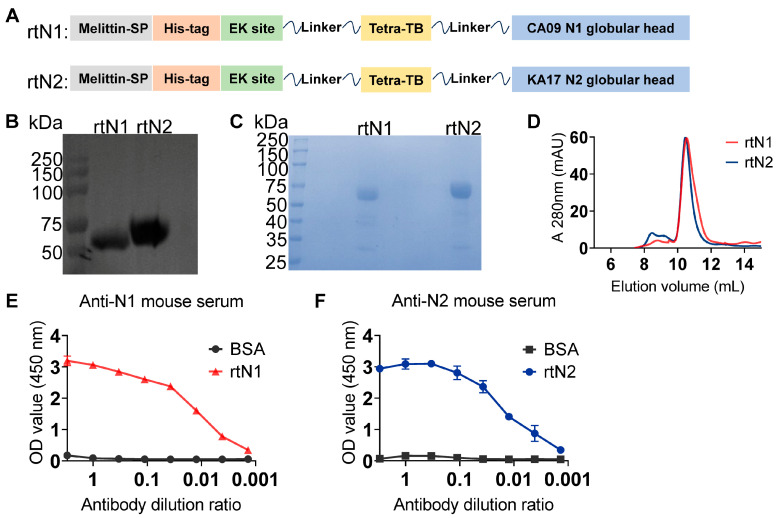
Construction and characterization of rtN1 and rtN2. (**A**) The diagram of protein construction depicts the compositions of rtN1 and rtN2. (**B**,**C**) Western blot and Coomassie blue staining analyses of rtN1 and rtN2. The hexahistidine-specific mAb was used for blotting rtNA protein bands. (**D**) SEC analysis of rtN1 and rtN2 proteins. The rtNA samples were collected at the major peak between the elution volumes of 10 and 11 for later protein characterization and immunization. (**E**,**F**) Antigenicity analysis of rtN1 and rtN2 using the ELISA method. Immunoplates coated with bovine serum albumin (BSA) were used as the negative control. Error bars represent the standard derivations (SDs) of the values of triplicate samples (*n* = 3 per group).

**Figure 2 viruses-16-00077-f002:**
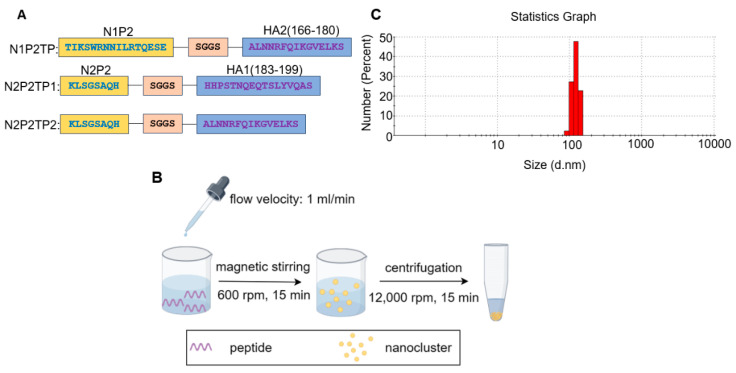
Preparation and characterization of peptide nanoclusters. (**A**) The peptide construction diagram depicts the amino acid sequences of composite peptides. N1P2TP peptide consists of N1P2, HA2 (166–180), and a SGGS linker in between; N2P2TP1 peptide consists of N2P2, HA1(183–199), and a SGGS linker in between; and N2P2TP2 peptide consists of N2P2, HA2 (166–180), and a SGGS linker in between. (**B**) The diagram depicts the preparation process of desolvated peptide nanoclusters. (**C**) Size-distribution analysis of N2P2TP2 nanoclusters based on DLS.

**Figure 3 viruses-16-00077-f003:**
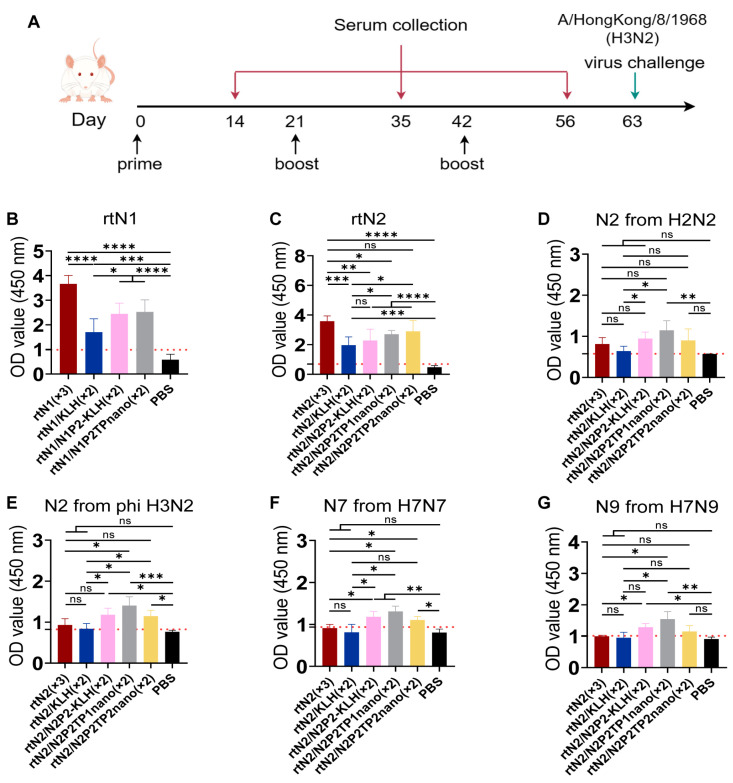
The schematic diagram of the immunization regimen and serum cross-reactivity analysis. (**A**) The schematic diagram depicts the time schedule of immunization, bleeding, and viral infection. (**B**–**G**) Serologic analysis. The hyper-immune serum samples collected on day 56 from each group (*n* = 6) were pooled and tested in triplicate using ELISA to evaluate serum reactivities to homologous (**B**) rtN1 and (**C**) rtN2 proteins, and (**D**–**G**) NA proteins from other strains expressed and displayed on the plasma membrane of transiently transfected HEK293T cells, (**D**) A/Japan/305/1957 H2N2, (**E**) A/Philippines/2/1982 (Phi) H3N2, (**F**) N7 from A/Netherlands/219/2003 H7N7, and (**G**) N9 from A/Shanghai/02/2013 H7N9. Error bars represent the SD of the results of triplicate samples. The red dash lines represent the values of mean + 2 × SD from the PBS group. *, **, ***, and **** indicate significant difference between the compared groups and represent *p*-values less than 0.05, 0.01, 0.005, and 0.001, respectively. The abbreviation ns indicates not significant.

**Figure 4 viruses-16-00077-f004:**
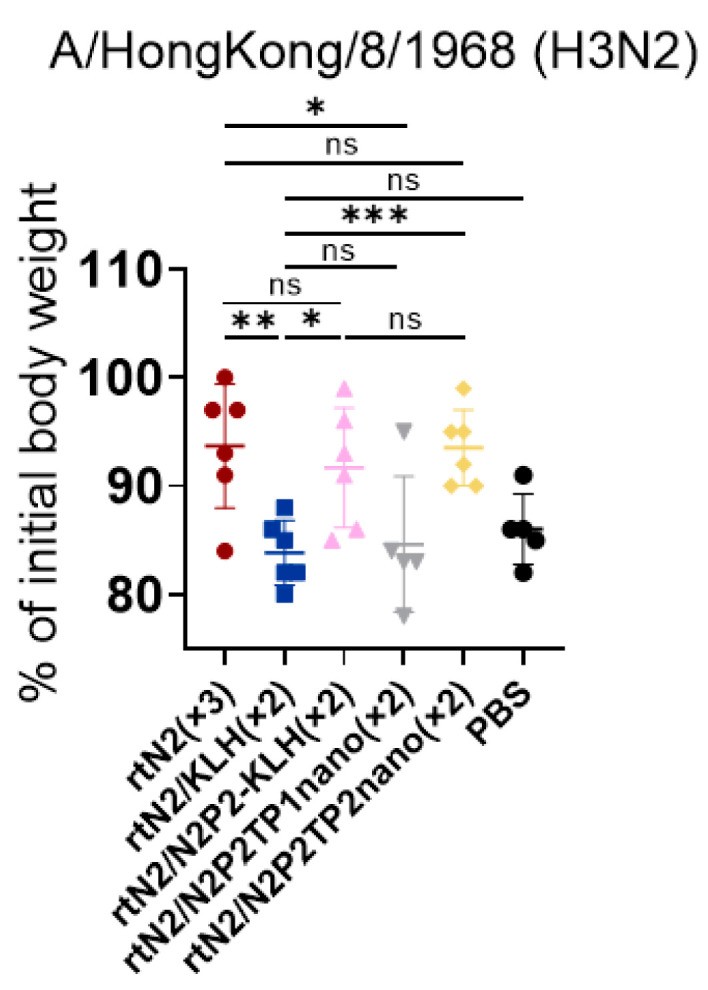
Mouse challenge experiment with HK68 H3N2 virus. Body weights from five groups of mice with N2 immunities were recorded on the day of intranasal infection and on the 5th day post infection. Statistical significance between the compared groups was determined using an unpaired *t*-test. *, **, and *** indicate significant difference between the compared groups and represent *p*-values less than 0.05, 0.01, and 0.005, respectively. The abbreviation ns indicates not significant.

**Figure 5 viruses-16-00077-f005:**
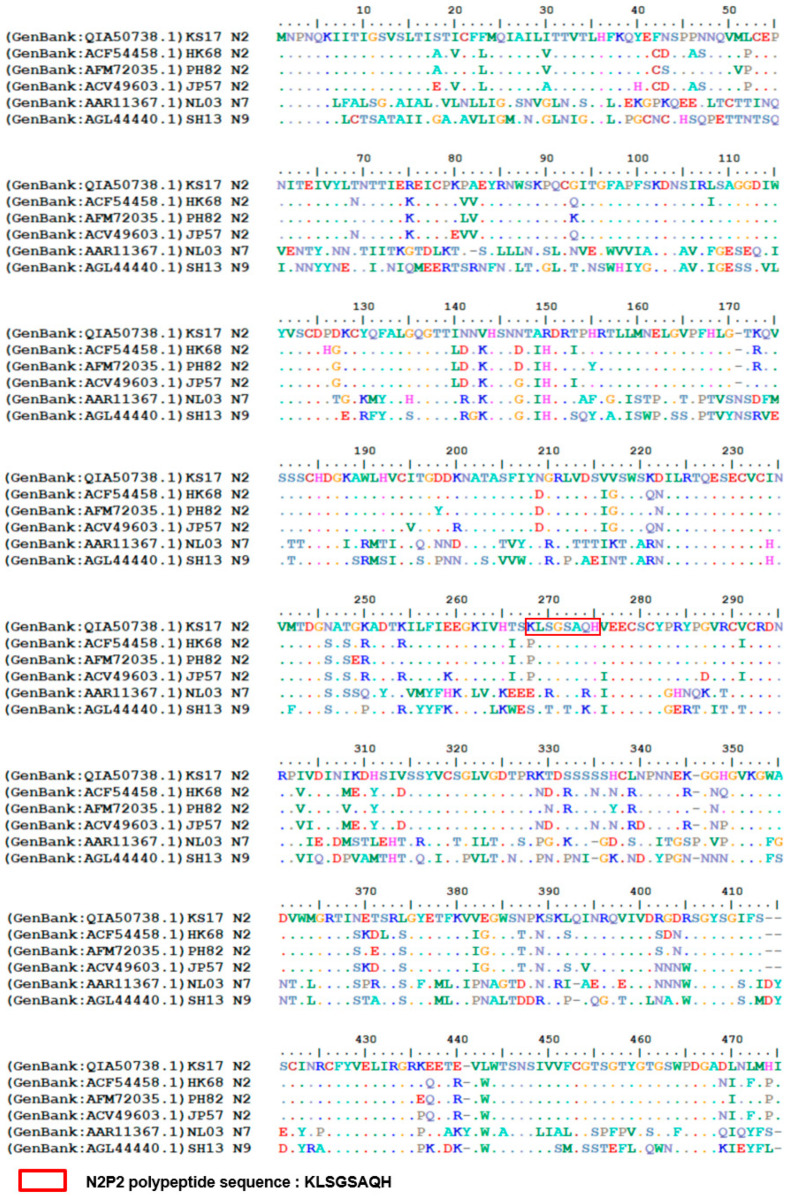
The alignment of NA amino acid sequences. The N2 head domain of rtN2 immunogen is derived from KS17 N2. The amino acid sequence of N2P2 is selected based on the consensus amino acid sequence of N2 [8] and is also identical to the corresponding peptide from KS17 N2. The HK68 H3N2 virus was used in the mouse challenge experiment. The eukaryotic expression plasmids encoding NA from PH82, JP57, NL03, and SH13 were used for transfection in cell-based ELISA.

## Data Availability

Further information and requests for resources and reagents should be directed to and will be fulfilled by the lead contact Lei Deng (ldeng@hnu.edu.cn).

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
