# Peer review of "Sequential Immunizations with Influenza Neuraminidase Protein Followed by Peptide Nanoclusters Induce Heterologous Protection"

_viruses, 2024, doi:10.3390/v16010077_

Round 1

Reviewer 1 Report

Comments and Suggestions for Authors

The paper is extremely interesting and could be useful in the future. According to this referee, some essential experimental controls are obviously needed for publication

1) Do you have evidence that the immunity response obtained with female mice is also comparable to that of male mice?

2) The authors should better specify in the text from which influenza strains the rtN1 and rtN2 peptides were cloned.

3) The article does not mention any of the reasons for choosing to use 3 ug of N1 or N2 in nanoclusters. Has dose-response been done previously? Can the data also be shown?  Also, there is no mention of the DL50 dose used, which should be included in the materials and methods section.

4) As an experimental reading, the authors checked only the weight of the mice and the antibody/cross response. There is no evidence to other data, such as clinical score, survival of mice, and especially how this type of vaccination can reduce viral infection. For example, the viral titer after immunization should absolutely be evaluated. Similarly, consideration should be given to whether immunization with nanoclusters can give adverse reactions, thus evaluating lung tissue (histopathology) or analyzing cytokines from BAL or plasma.

5) The application of nanoparticles for vaccine preparation should be discussed in more detail in the discussion. Are these preparations being tested? Or on the market available? Have they been developed so far only for vaccines against viral antigen or also bacteria antigen?

Author Response

Responses to Reviewer 1 Comments

The paper is extremely interesting and could be useful in the future. According to this referee, some essential experimental controls are obviously needed for publication

1) Do you have evidence that the immunity response obtained with female mice is also comparable to that of male mice?

Response: The objective of our experiments is to evaluate the immunogenicity of NA peptide nanoclusters for effectively boosting NA protein-specific antibody response. Humoral immune responses to immunization and infection are generally stronger in female than male. It is usual to utilize the female mouse model for vaccine immunogenicity evaluation.

2) The authors should better specify in the text from which influenza strains the rtN1 and rtN2 peptides were cloned.

Response: Thanks for reviewer’s suggestion. The amino acid sequences of NA head domains in rtN1 and rtN2 are derived from NA sequences of influenza A trains A/California/04-CIP100_RGCM_1868/2009 (H1N1) N1 (GenBank Protein Accession: AWZ60965.1) and A/Kansas/14/2017 (H3N2) N2 (GenBank Protein Accession: QIA50738.1), respectively. We have made the correction on the lines 92 - 96 in the revised manuscript.

3) The article does not mention any of the reasons for choosing to use 3 ug of N1 or N2 in nanoclusters. Has dose-response been done previously? Can the data also be shown?  Also, there is no mention of the DL50 dose used, which should be included in the materials and methods section.

Response: Prime immunization with 3 μg rtNA works in the elicitation of NA protein-specific memory B cell response. In this study, we just used the same dose of NA protein as in the previous study for priming. The challenge dose of 6 × LD50 was used this time, as mentioned in the materials and methods. We have titrated the used HK68 H3N2 virus previously, the actual dose of 1 × LD50 contains 100 plaque forming units of HK68 H3N2 virus. We have explained this on the line 144 in the revised manuscript.

4) As an experimental reading, the authors checked only the weight of the mice and the antibody/cross response. There is no evidence to other data, such as clinical score, survival of mice, and especially how this type of vaccination can reduce viral infection. For example, the viral titer after immunization should absolutely be evaluated. Similarly, consideration should be given to whether immunization with nanoclusters can give adverse reactions, thus evaluating lung tissue (histopathology) or analyzing cytokines from BAL or plasma.

Response: Thanks for reviewer’s suggestion. Recording body weight loss of mice after infection is one of the reliable ways to reflect vaccine protectiveness. The clear conclusion can be drawn from the result of Figure 4, that N2P2TP2nano boosters confer as slower body weight drop as rtN2(×3) group. We think the H3N2 virus inocula for mice infections were dosed too heavily in the challenge experiment, all mice died in the period from day 7 to day 10 after infection.

5) The application of nanoparticles for vaccine preparation should be discussed in more detail in the discussion. Are these preparations being tested? Or on the market available? Have they been developed so far only for vaccines against viral antigen or also bacteria antigen?

Response: Thanks for reviewer’s suggestion. We have introduced the marketed anti-cancer drug products that are developed based on the desolvation nanoparticle technology on the lines 295 – 302 in the revised manuscript.

Reviewer 2 Report

Comments and Suggestions for Authors

This is a nicely written manuscript that describes development of an immuno focusing method using nanoclusters to enhance response to cross-reactive B cell epitopes in the neuraminidase (NA) protein.  The nanoclusters do not involve carrier protein and consist only of peptides, a B cell NA epitope and a T cell helper epitope from HA.   Mice primed with recombinant NA protein and boosted with N2 nanoclusters demonstrated greater cross-reactivity in cell-based ELISA than recombinant protein alone.  A heterologous H3N2 challenge in mice was conducted to show protection based on weight loss on day 5 post-challenge.

Overall, the presented data support that nanoclusters enhance immunogenicity against cross-reactive epitopes.  The challenge/protection data is weaker.

Minor considerations to improve the manuscript:

·       Line 23- Please define rt.

·       Section 2.4, lines 113-114 .  Please clarify H3 peptide helper epitopes used.  It is not clear that two different helper epitopes are used as shown in Figure 2a.

·       Challenge data appears incomplete and not necessarily in line with immunogenicity data.  The HK/68 virus is a lethal model as 6xmLD50 dose is used.  However, no data on survival is given.  Did any of the naïve mice die?  The weight loss data on a single timepoint is incomplete picture of challenge.    Challenge virus titer in lungs would also be more indicative of the protection.   

·       Can a summary of the homology between the HK/68 NA and Kansas2017 NA be provided to demonstrate to the reader that the one epitope is conserved and that the two NA sequences are divergent?

·       N2P2TP1 performs better than helper epitope from HA2. Why?

·       Figure 3 figure legend , lines 251-256 -please clarify whether panels D through G are recombinant N2 proteins of cell-based Elisa.  Implication is recombinant proteins.

·       No baseline serum titers are shown or mentioned.  Were all mice groups similar at baseline?  Pre-immunization titers for at least homologous protein would be useful to understand the day 56 titers.

Author Response

Responses to Reviewer 2 Comments

This is a nicely written manuscript that describes development of an immuno focusing method using nanoclusters to enhance response to cross-reactive B cell epitopes in the neuraminidase (NA) protein.  The nanoclusters do not involve carrier protein and consist only of peptides, a B cell NA epitope and a T cell helper epitope from HA.   Mice primed with recombinant NA protein and boosted with N2 nanoclusters demonstrated greater cross-reactivity in cell-based ELISA than recombinant protein alone.  A heterologous H3N2 challenge in mice was conducted to show protection based on weight loss on day 5 post-challenge.

Overall, the presented data support that nanoclusters enhance immunogenicity against cross-reactive epitopes.  The challenge/protection data is weaker.

Minor considerations to improve the manuscript:

  • Line 23- Please define rt.

Response: We have defined rt in the Abstract section on the line 23 in the revised manuscript.

  • Section 2.4, lines 113-114.  Please clarify H3 peptide helper epitopes used.  It is not clear that two different helper epitopes are used as shown in Figure 2a.

Response: Thanks for the reviewer’s suggestion. We have explained the different peptides in 2.4 section on the lines 117 - 118 in the revised manuscript.

  • Challenge data appears incomplete and not necessarily in line with immunogenicity data.  The HK/68 virus is a lethal model as 6xmLD50 dose is used.  However, no data on survival is given.  Did any of the naïve mice die?  The weight loss data on a single timepoint is incomplete picture of challenge.    Challenge virus titer in lungs would also be more indicative of the protection.

Response: Thanks for reviewer’s suggestion. Recording body weight loss of mice after infection is one of the reliable ways to reflect vaccine protectiveness. The clear conclusion can be drawn from the result of Figure 4, that N2P2TP2nano boosters confer as slower body weight drop as rtN2(×3) group. We think the H3N2 virus inocula for mice infections were dosed too heavily in the challenge experiment, all mice died in the period from day 7 to day 10 after infection.

  • Can a summary of the homology between the HK/68 NA and Kansas2017 NA be provided to demonstrate to the reader that the one epitope is conserved and that the two NA sequences are divergent?

Response: Thanks for the reviewer’s suggestion. We have aligned the amino acid sequences of the NA proteins from the strain A/Kansas/14/2017 (KS17, H3N2) (GenBank Protein Accession: QIA50738.1), A/Hong Kong/8/1968 (HK68, H3N2) (GenBank Protein Accession: ACF54458.1), A/Philippines/2/1982 (PH82, H3N2) (GenBank Protein Accession: AFM72035.1), A/Japan/305/1957 (JP57, H2N2) (GenBank Protein Accession: ACV49603.1), A/Netherlands/219/2003 (NL03, H7N7) (GenBank Protein Accession: AAR11367), and A/Shanghai/02/2013 (SH13, H7N9) (GenBank Protein Accession: AGL44440). The amino acid sequence of N2P2 (KLSGSAQH) is indicated below the alignment. The potential correlation of the variations in amino acid sequence with the immunogenicity impairment is analyzed and discussed in the last paragraph in Discussion section on lines 335 - 344 in the revised manuscript.

  • N2P2TP1 performs better than helper epitope from HA2. Why?

Response: As the OD values of serum binding activities to heterologous N2 and hetersubtypic N7 and N9 in the rtN2/N2P2TP1nano(×2) group showed even higher than the rtN2/N2P2TP1nano(×2) group in Figure 3, we think the incorporation of HA1(183-199) in composite peptide N2P2TP1 should has worked in helper responses and been even more effective than HA2(166-180) peptide possibly. Both T cell epitopes for incorporation must have provided adequate assistance in inducing NA peptide-specific antibody responses, HA1(183-199) inducing stronger T helper cell response. The severer body weight loss was recorded in the group rtN2/N2P2TP1nano(×2) at the 5th day after infection, implying the weaker protection conferred by N2P2TP1 peptide. For now, we cannot clearly explain this result.

  • Figure 3 figure legend , lines 251-256 -please clarify whether panels D through G are recombinant N2 proteins of cell-based Elisa.  Implication is recombinant proteins.

Response: Thanks for the reviewer’s suggestion. We have made the correction on the lines 242 - 243 in the revised manuscript.

  • No baseline serum titers are shown or mentioned.  Were all mice groups similar at baseline?  Pre-immunization titers for at least homologous protein would be useful to understand the day 56 titers.

Response: We have made changes in Figure 3, please check.

Round 2

Reviewer 1 Report

Comments and Suggestions for Authors

Please add in the text the response to question 1 with the references about it Response: The objective of our experiments is to evaluate the immunogenicity of NA peptide nanoclusters for effectively boosting NA protein-specific antibody response. Humoral immune responses to immunization and infection are generally stronger in female than male. It is usual to utilize the female mouse model for vaccine immunogenicity evaluation.

Author Response

The reviewer's comment:

Please add in the text the response to question 1 with the references about it Response: The objective of our experiments is to evaluate the immunogenicity of NA peptide nanoclusters for effectively boosting NA protein-specific antibody response. Humoral immune responses to immunization and infection are generally stronger in female than male. It is usual to utilize the female mouse model for vaccine immunogenicity evaluation.

Response: Please check these two papers published in recent years, where the phenomenon that stronger specific humoral immune response can be induced in female than male is well explained. 

Reference 1: Zhao, Ruozhu et al., A GPR174-CCL21 module imparts sexual dimorphism to humoral immunity. Nature, 2020, 577,7790.

Reference 2: Sabra L. Klein et al., Sex differences in immune responses. Nature Reviews Immunology, 2016, 16, 626–638.